# Iris Colour and the Risk of Developing Uveal Melanoma

**DOI:** 10.3390/ijms21197172

**Published:** 2020-09-28

**Authors:** Laurien E. Houtzagers, Annemijn P. A. Wierenga, Aleid A. M. Ruys, Gregorius P. M. Luyten, Martine J. Jager

**Affiliations:** Department of Ophthalmology, Leiden University Medical Center, Albinusdreef 2, 2333 ZA Leiden, The Netherlands; A.P.A.Wierenga@lumc.nl (A.P.A.W.); aleidruys@gmail.com (A.A.M.R.); g.p.m.luyten@lumc.nl (G.P.M.L.)

**Keywords:** eye diseases, oncology, uveal melanoma, iris colour, epidemiology, melanin

## Abstract

Uveal melanoma (UM) is a global disease which especially occurs in elderly people. Its incidence varies widely between populations, with the highest incidence among Caucasians, and a South-to-North increase in Europe. As northern Europeans often have blond hair and light eyes, we wondered whether iris colour may be a predisposing factor for UM and if so, why. We compared the distribution of iris colour between Dutch UM patients and healthy Dutch controls, using data from the Rotterdam Study (RS), and reviewed the literature regarding iris colour. We describe molecular mechanisms that might explain the observed associations. When comparing a group of Dutch UM patients with controls, we observed that individuals from Caucasian ancestry with a green/hazel iris colour (Odds Ratio (OR) = 3.64, 95% Confidence Interval (CI) 2.57–5.14) and individuals with a blue/grey iris colour (OR = 1.38, 95% CI 1.04–1.82) had a significantly higher crude risk of UM than those with brown eyes. According to the literature, this may be due to a difference in the function of pheomelanin (associated with a light iris colour) and eumelanin (associated with a brown iris colour). The combination of light-induced stress and aging may affect pheomelanin-carrying melanocytes in a different way than eumelanin-carrying melanocytes, increasing the risk of developing a malignancy.

## 1. Introduction

The most common primary malignant tumour in the eye in adults is uveal melanoma (UM), which is derived from mutated ocular melanocytes [1]. UM can be divided into posterior and anterior tumours, with the posterior ones located in the choroid and the ciliary body, the anterior ones in the iris [2]. Together, these areas are known as the uvea or uveal tract [3]. The majority of UM (90%) incidences develop in the choroid. Melanomas originating in the ciliary body are less common (6%), and development in the iris is the least common (4%) [4]. The disease occurs more frequently in countries with a population of Caucasian descent and is especially common in northern European countries [5].

Many patients visit the doctor rather late because of a lack of symptoms. When patients do have symptoms, they mostly present with blurred vision, photopsia (seeing flashes of light), visual field loss or a visible tumour. Older patients are more likely to report no symptoms [6]. Most tumours can be treated by irradiation (radioactive plaque, proton beam), while large tumours may necessitate enucleation of the eye [7]. UM is associated with a 50% risk of metastases, predominantly to the liver [8,9]. Risk factors for a poor prognosis of UM are large tumour size, ciliary body location and an epithelioid cell type [4,10,11]. Currently, treatment of metastases is often not curative, but liver perfusion or surgical resection of metastases may prolong life. In iris melanoma, patients may mention a change in iris colour or in the shape of the pupil [12,13].

Understanding the biological basis of UM may help to identify treatment targets. We determined whether eye colour differed between Dutch UM patients and Dutch population controls. We confirmed that a light iris colour is a predisposing factor for the development of UM in a population of Caucasian ancestry. As melanin is the pigment that determines iris colour, we investigated the relationship between the different types of melanin and UM. This may help to gain more insight into the pathological role of melanin in the development of UM.

## 2. Iris Colour Distribution in The Netherlands

In The Netherlands, the majority of the population is Caucasian. While many Dutch have blue eyes, we wondered whether Dutch UM patients would display the same distribution in eye colour in comparison to the general Dutch population. We compared the distribution of iris colour between UM patients and the normal Dutch population, asking the question whether a lighter iris colour in the Dutch population is associated with a greater risk of UM.

In 2010, a study took place regarding the distribution of iris colour in Rotterdam, The Netherlands. Rotterdam and Leiden are both cities in Zuid-Holland, the most populated Dutch province, where people originating from all the provinces of The Netherlands move to because it is the economic centre. Therefore, it is assumed that the sample from Rotterdam is a representative sample of iris colour for the Dutch population. The Rotterdam Study (RS) consists of three different population cohorts, RS1 started in 1990, RS2 started in 1999 and RS3 in 2006. Eventually, the iris colour of 5591 Dutch Europeans was determined from high-resolution digital full-eye photographs, using a Sony HAD 3CCD colour video camera with a resolution of 800 × 600 pixels (Sony Electronics Inc., New York, NY, USA), identifying three different types of iris colour: blue, green-hazel/intermediate and brown. Two independent researchers reviewed the iris colours of the 5591 participants, of whom both genotypic information and eye photos were available. In this study, 69.5% of the participants had blue eyes, 7.7% an intermediate phenotype and 22.8% had brown eyes [14].

## 3. Research Design and Methods

### 3.1. Study Approval

This research was approved by the Biobank of the LUMC (Leiden University Medical Center) (number: Uveamelanoomlab-2019-8 METC B20.024). This research adhered to Dutch law and the tenets of the Declaration of Helsinki (World Medical Association of Declaration 1964; ethical principles for medical research involving human subjects).

### 3.2. Study Population

#### Leiden Cohort

Using a database consisting of 1216 UM patients who underwent enucleation for UM at the LUMC, Leiden, The Netherlands, a total of 412 UM patients was selected of whom we had data on iris colour. The iris colour was obtained from patient’s medical charts as well as from clinical photographs. Iris colours were divided in three different groups: blue/grey, intermediate and brown. The intermediate iris colour consisted of eyes with green and hazel colours.

## 4. Results

In Table 1, the baseline characteristics of the study population of the Rotterdam Study (RS123) (n = 5951) and the Leiden UM cohort (n = 412) are shown. The male-to-female ratio of both populations was significantly different (*p* < 0.001), which is probably due to the fact that more men than women with UM had their eye colour described in the database. The mean ages of the two groups were comparable, with a mean age of 63.0 in the Leiden Cohort and 66.4 in RS123. There was a significant difference in the distribution of iris colour between the two groups (*p* < 0.001). In both populations, blue or grey eyes were the most frequent: 70% of the RS123 cohort had blue/grey eyes compared with 65% of the UM patients. Brown iris colour was more frequent in the RS123 cohort (23%) than in the Leiden UM patient population (16%). The odds ratio for getting UM in people with a blue/grey iris colour was 1.38 (95% CI 1.04–1.82) in comparison to brown eyes. In the UM patient group, an especially large part of the population had a green or hazel iris colour (19%) in comparison to the RS123 cohort (8%). Individuals with green/hazel eyes were found to have a significantly higher crude risk of UM than those with brown eyes (OR = 3.64, 95% CI 2.57–5.14).

We compared our findings with other studies that have published a cohort of UM patients with known iris colours in Table 2 and Table 3. Some studies used the same iris colour categories as we used in our study (brown, blue/grey and green/hazel), while other studies provided each iris colour as a separate category (brown, blue, grey, green and hazel), allowing us to place them into the three categories. Two studies originated from Canada, two from Australia, two from Germany and one from France. Three of these studies investigated the iris colour distribution of UM patients, three of “ocular melanoma” and one of “choroidal plus ciliary melanoma”. All studies, except for Rootman et al. [15], had age- and sex-matched controls. The size of our control group (n = 5,951) was much larger than in any of the other studies, and the number of UM patients in our study was also relatively large. The odds ratios from the other studies are all crude odds ratios, so they are not adjusted for age, gender, region or other host factors.

If we compare our findings with the other studies in Table 2, the distribution of iris colour among the cases seems comparable. The data from the study of Stang et al. [20] are most similar to our population, with 64% of their control population having a blue/grey iris colour, 11% a green/hazel iris colour and 24% a brown iris. The controls in this study came from Essen, which is close to the Dutch border. In all studies, a remarkably low percentage of patients have brown eyes.

In all the studies, the odds ratios for blue/grey and the odds ratio for green/hazel compared to brown were higher than 1, although not all of them were significantly higher, as shown in Table 3. The odds ratio for green/hazel from our study was relatively high and the odds ratio for blue/grey was relatively low in comparison to the other studies.

## 5. Incidence

Most people who get a UM are between 50 and 70 years old, but this tumour occasionally develops in younger people [22]. Males have a slightly higher incidence of UM than females [23]. The reported 5-year survival rate ranges from 66% to 82% [24,25,26,27].

UM is a rare disease: it occurs in approximately two to eight cases per million people per year in the United States of America and Europe [23,24,28,29]. Asian or African ancestry is uncommon, with an incidence rate of 0.42 per million per year in South Korea [30,31,32]; UM in Asians has a slightly different clinical presentation, as patients from Asia present with UM at a younger age, with an average of 42.9 years in South East Asia versus an average of 60.4 years in Western countries [33,34]. Patients with an African ancestry are more likely to show secondary glaucoma [35]. Because of the relation with Europe, an association has been described with light hair, a light skin colour and light iris colours [16,18,36,37]. In 2007, the EUROCARE (European Cancer Registry) workgroup investigated the distribution of 6673 UM patients diagnosed between 1983 and 1994 in 16 European countries with different latitudes [24]. In Spain and in southern Italy, the incidence rate of UM was less than two cases per million, whereas in Norway and Denmark the incidence rate was more than eight cases per million [24]. Per 10° increase of latitude, the increase in the incidence rate ratio was 1.40 (95% confidence interval (CI) 1.09–1.80) [24]. Incidence rates remained stable during the study period. The Netherlands was also involved in this study: according to the Eindhoven Cancer Registry, with 49 cases of UM, the incidence rate was 4.8 per million (95% CI 3.5–6.2) [24]. Remarkably, in 2019, an observational study was published about the incidence of UM in Ireland. Ireland had not been included in the study by the EUROCARE workgroup, and the mean age-adjusted incidence of UM in Ireland was 9.5 per million, which is higher than in any of the countries involved in the EUROCARE study [38]. An explanation could be that risk factors of UM such as light skin colour, light hair colour and light eye colour are stereotypical traits of the native Irish population [16,18,36,37].

In the USA, the incidence rate is 5.2 per million, which has remained almost stable between 1973 and 2013, with only a 0.5% annual increase among white patients. In the same time period, no change occurred in the 5-year relative survival rate of 81%, despite changes in the primary treatment of UM [26]. Northern European ancestry was observed to be a risk factor for UM in a study in the USA [5,28].

## 6. Origin

Both UM and cutaneous melanoma originate from melanocytes [36]. Between 2008 and 2012, for cutaneous melanoma, a 5-year relative survival of 89% was found in all age groups in The Netherlands. Between 1989 and 2008, the average population size of The Netherlands was 15.7 million. During this 20-year period, 19,393 males and 26,526 females were diagnosed with cutaneous melanoma, and 5840 males and 4,769 females died because of this malignancy. While the incidence rate of cutaneous melanoma increased by 4.1% annually, so did the 10-year survival rate: for males, the 10-year relative survival improved significantly from 70% in 1989–1993 to 77% in 2004–2008 (*p* < 0.001) and for females from 85% to 88% (*p* < 0.01). An earlier diagnosis was associated with a better prognosis [39]. Neither this increase in incidence nor an improvement in survival has been observed in UM [40].

Although UM and cutaneous melanoma both originate from melanocytes, their biological behaviour is different, as well as the driver mutations. UMs most often have a *GNAQ* or *GNA11* mutation, while cutaneous melanomas usually have a *BRAF*, *NRAS*, *KIT* or *NF1* mutation [41]. However, the same *BRAF* mutation as in cutaneous melanoma has been found in some iris melanomas and in a fraction of cells of some posterior UMs [42,43,44]. UM metastasizes mostly via the blood to the liver, whereas cutaneous melanoma spreads to the lungs, liver, brain and bones via the lymphatic system. Immunotherapy such as anti-PD1 (programmed cell death protein 1) anti-PD-L1 (programmed cell death protein—ligand 1) or CTLA-4 (cytotoxic T-lymphocyte-associated protein 4) therapy is often applied successfully in cutaneous melanoma, but has limited success in UM [45]. Rare UM with specific germline *MBD4* mutations has been described to respond to anti-PD-1 therapy, probably because *MBD4* mutations are associated with a high tumour mutation burden [46,47].

## 7. Risk Factors

Some people have a greater risk of developing UM than others, and knowledge of the pathophysiology behind this may aid in explaining UM and finding a cure. Some of the risk factors include a light iris colour, northern European heritage, age (the older, the greater the risk), gender (UM is slightly more common in men than in women), a high numbers of freckles, a lack in the ability to tan, a high number of moles and a previous history of cutaneous melanoma [5,18,23,37,48,49]. Cutaneous melanoma, which also develops from mutated melanocytes, is highly correlated with sun exposure, although the relationship between ultraviolet radiation (UV) and cutaneous melanoma is not yet fully understood [50,51]. However, the situation is more complicated for UM: different factors like lifestyle, living in urban or rural areas, geographic location and cumulative life exposure to intense sunlight do play a role, but sunlight is thought to be less important than in cutaneous melanoma [5]. It is noteworthy that while, through time, an increase in the incidence of cutaneous melanoma has occurred in parallel with more sun exposure, no such increase has been observed in UM [1,5,23,52]. A meta-analysis from Nayman et al., looked at the role of time spent on outdoor leisure activities, occupational sunlight exposure and the latitude at birth in relation to developing UM [37]. None of these parameters were found to be significant risk factors. However, welders were found to have a significantly higher risk of getting a UM (odds ratio of 7.3 with a 95% CI between 2.6 and 20.1 for men). In this case, exposure to UV light could be a causal agent, but the higher risk could also be explained by other factors associated with welding [18,37]. Use of sunlamps has also been significantly correlated with UM [5]. No significant relationship between mobile phone use and UM has been found [53].

A genetic risk factor is the presence of a germline mutation in the *BAP1* gene, which is associated with the *BAP1*-tumor predisposition syndrome. Patients with a mutation in this gene have an increased chance of developing UM, melanocytic cutaneous tumours, mesothelioma and renal cell carcinoma [54,55,56]. The presence of a Naevus of Ota, or congenital ocular melanocytosis, constitutes a pigment disorder which carries an increased risk of developing UM [57,58].

Multiple studies in Canada, the USA, Germany, France and Australia have shown that UM has a higher incidence among people with a light iris colour [5,16,17,18,20,59]. Other studies argue more towards an association with skin type [5], but it is hard to separate the two. Furthermore, an association between the risk of metastatic death from UM and blue or grey irises has been found [60]. No statistical difference in metastasis or death from UM based on ethnic background has been reported [61].

Some researchers suggest that the reason that a dark iris colour leads to a lower chance of developing UM than a light iris colour is because there is more protection from UV radiation due to the higher percentage of melanin in dark eyes [29,62]. However, UV hardly penetrates to the back of the eye. It does however reach the most frontal part of the uvea, the iris. The correlation between iris colour and iris melanoma is more pronounced than for posterior UM. In a cohort studied by Rootman and Gallagher, 21 out of the 23 patients with iris melanoma had blue or grey eyes, and none of them had brown eyes. The colour distribution was significantly different from the control group and the population with posterior UM [15].

## 8. Tumour Development

It is known that posterior UMs carry specific mutations: in 94% of all tumours, a mutation in either the *GNAQ* or *GNA11* gene has been found [63]. These mutations are also found in choroidal nevi [64]. *GNAQ* and *GNA11* occur in a mutually-exclusive fashion, in about equal numbers [28,41,65]. The posterior choroid is the ocular region that is most exposed to focused visible light and it is plausible that damaging visible light is involved in the development of UM in this area. From the whole visible light spectrum, short-wave blue light with high energy is probably the most harmful for the eye and most likely to cause UM [66]. As the cornea, lens and vitreous filter out most UV, almost no UV light reaches the retina, but as in welding, non-UV wavelengths may play a role in oncogenesis [50,67]. Tumours located in the transition zone between the ciliary body and choroid receive no visible light. Among all UM, these are most associated with light-coloured eyes and they often have an A > T mutation in *GNAQ* Q209L or *GNA11* Q209L. This is not the typical C > T mutation signature associated with UV light. This suggests that the risk of developing a UM in light eyes is independent of light exposure [63].

Other common genetic aberrations in posterior UM are chromosomal changes, such as a loss of chromosome 3, loss of chromosome 8 p and a gain of chromosome 8 q, which are associated with a higher risk of developing metastases [68,69,70]. Whereas *GNAQ* and *GNA11* mutations occur early in tumour formation and are not associated with prognosis, *EIF1AX*, *SF3B1* and *BAP1* mutations occur later in the development of the tumour and are related to prognosis. A mutation in *EIF1AX* is present in 17% of primary UM, a mutation in *SF3B1* in 25% and a mutation in *BAP1* in about 45% of primary UM [71]. These three mutations are almost always mutually exclusive [41,72]. *EIF1AX* mutations are associated with a good prognostic outcome; tumours with an SF3B1 mutation lead to an intermediate prognosis, in contrast to UMs with a *BAP1* mutation. *BAP1*-mutated tumours are strongly associated with metastases and have a high melanoma-specific mortality [72,73,74]. The *EIF1AX* mutation is responsible for unsuccessful protein translation, and the SF3B1 mutation affects gene splicing [71]. The lack of the *BAP1* protein interferes with a wide range of normal cell processes such as DNA damage repair; 40% of UM metastases have a *BAP1* mutation [75].

Iris melanomas show a different mutation pattern. *GNAQ* and *GNA11* mutations are detected in 77% to 84% of the iris tumours, which are not as frequent as in UM [63,76,77]. *GNAQ* and *GNA11* are mutually exclusive in iris melanomas, and the mutation status of *GNAQ* and *GNA11* does, similar to UM, not correlate with patient survival [76,77,78]. More iris melanomas have a *GNAQ* mutation (47%) than a *GNA11* mutation (30%) [76]. As in UM, mutations in *BAP1*, *EIF1AX* and *SF3B1* are frequently seen in iris tumours [76]. The frequency of *BAP1* mutations seems comparable with UM. However, in iris tumours, the *BAP1* mutation status has no correlation with patient survival [76]. Although not much data about these mutations are available, *EIF1AX* mutations seem to occur more often than in posterior UM [76,77]. As *EIF1AX* mutations are correlated with a good prognosis in UM, this could be one of the explanations for the relatively good survival of patients with an iris tumour in comparison to more posteriorly located UM [79]. *SF3B1* mutations are less common in iris tumours [77]. In a study by Van Poppelen et al., 10 out of 30 iris tumours had mutations in *NRAS*, *BRAF, PTEN*, *c-KIT* and/or TP53 [76]. These mutations are not detected in UM. Another study on 19 cases showed a *BRAF* mutation in 47% of the iris tumours [43]. As previously mentioned, mutations in BRAF and NRAS are common in cutaneous melanoma [41]. Seventy-one percent of iris tumours have a complete or partial loss of chromosome 3, and some tumours have a chromosome 9 loss [80,81]. Iris melanomas have a relatively high mutation burden compared to other UMs, which is driven by UV radiation [82]. Immunotherapy, which is helpful in UMs with a *MBD4* mutation and a high mutation burden, is therefore probably also useful for iris melanomas, but luckily, these tumours do not often give rise to metastases.

## 9. Iris Colour

In order to reach the choroid, light has to penetrate multiple tissues and humours: the cornea, the aqueous humour, the lens, the vitreous (humour) and the retina. The retina involves the retinal pigment epithelium (RPE) and de neural retina [83]. UV light can be divided into three categories: ultraviolet radiation (UV) A with a wavelength between 320 and 400 nm, UVB with a wavelength between 280 and 320 nm and UVC with a wavelength between 100 and 280 nm [84]. The pigment epithelium and the melanocytes in the iris block and absorb both visible light and UV radiation [67,85], and light can therefore only enter through the pupil. However, as the cornea absorbs UVC and most UVB, and the lens and vitreous absorb almost all energy of wavelengths of nearly 400 nm, only visible light reaches the RPE [67,85]. Because UVA irradiation causes oxidation of pre-existing melanin and UVB causes direct DNA damage, both types of irradiation can be harmful and carcinogenic [86]. Recent data however indicate a contribution of UV radiation to DNA damage in UM. UV radiation may damage DNA in different ways, such as by creating the main promutagenic DNA adduct cyclobutene pyrimide dimer or through the formation of reactive oxygen species (ROS) [87,88]. In a recent study, the C > T substitute in DNA (associated with UV light) was even higher in UM than in cutaneous melanoma [44]. Because the iris is directly exposed to UV light, UV light may especially play an important role in the development of iris melanoma. Iris melanomas are mostly seen in the inferior part of the iris, where the exposure to sunlight is the most explicit. The choroid and the ciliary body are not directly exposed to sunlight. Exposure of the skin to sunlight is essentially positive, since it stimulates the synthesis of vitamin D [89]. As people in northern countries get less exposure to sunlight, this may affect vitamin D synthesis, which can be another explanation why there is a South-to-North increase of the incidence of UM [24].

One can identify two types of pigment cells in the eye: the retinal, iris and ciliary pigment epithelium, all of which originate from the neural ectoderm, and the uveal melanocytes, which originate from the neural crest, similar to skin and hair melanocytes [83,90]. Both uveal melanocytes and pigment epithelial cells can be isolated and cultured [91]. Uveal melanocytes maintain their production of melanin at a constant level whereas adult pigment epithelial cells do not produce melanin [91]. The amount of melanin in the RPE decreases over time, and the production of new melanosomes stops at the age of two [92,93]. Uveal melanin, especially in the ciliary body and choroid, can protect melanocytes by deactivating ROS, thereby reducing the chance of malignant transformation of uveal melanocytes [94].

The number of melanocytes in the iris does not differ per iris colour group (light, intermediate or dark iris colour) [95,96]. Asians have significantly fewer melanocytes [97]. As there is no difference in the number of melanocytes, the difference in iris colour is determined by the melanosome composition and structure of the melanocytes in the stroma [98].

There are two different types of melanin: eumelanin and pheomelanin [90]. The pigment epithelium mainly contains eumelanin. The main determinants of iris colour are the amount of eumelanin and the ratio of eumelanin to pheomelanin [99]. Uveal melanocytes can produce both eumelanin and pheomelanin in melanosomes [100]. An important regulator of the eumelanin/pheomelanin ratio is melanocyte-stimulating hormone (MSH), which binds to its receptor (MC1R) on melanocytes. This receptor is only present on melanocytes, the brain, active monocytes, macrophages and dendritic cells [101]. The activation of MC1R on melanocytes leads to an increase of the intracellular levels of cAMP [102]. Mutations of MC1R have been identified, of which the Arg151Cys, Arg160Trp and Asp294 variants have been reported to be over-represented in individuals with fair hair and skin, but they have not been associated with specific eye colours (see below) [103,104]. Li et al., studied the role of MC1R and MSH in uveal melanocytes [105]: human uveal melanocytes were derived from the choroid and the iris of deceased donors and then cultured [105] and exposed to different dosages of MSH. In human epidermal melanocytes, MSH leads to proliferation of the melanocytes, but proliferation was not observed in any of the uveal melanocyte cell lines, which was consistent with an earlier study [105,106]. The addition of MSH did not lead to higher levels of tyrosinase hydroxylase and tyrosinase on uveal melanocytes [105]. MSH did also not provide a higher level of Dihydroxyphenylalanine (DOPA) oxidase, in contrast to an earlier study conducted with human uveal melanocytes [105,107]. Uveal melanocytes do not express MC1R or MSH [105]. Therefore, MSH does not seem to play an important role in ocular pigmentation [105]. Furthermore, no difference in MC1R gene variant distribution has been observed between UM patients and controls [108]. This suggests that specific *MC1R* gene variants do not play a major role in the susceptibility to develop UM. However, as 95% of the primary UMs and 94% of the UM metastases express the MC1R receptor, it may be helpful as a target for new therapies [109,110].

The pathways leading to the production of eumelanin and pheomelanin have been studied extensively. An increase in intracellular cAMP leads to activation of tyrosinase, which leads to the oxidation of tyrosine into DOPA quinone [111], the precursor for eumelanin and pheomelanin. In the synthesis pathway of eumelanin, the addition of an amino group to DOPA quinone results in the formation of leucadopachrome [111]. Via a redox exchange, leucadopachrome is converted to dopachrome [95,111]. Dopachrome is a precursor for both 5,6-dihydroxindole [84], and carboxylated 5,6-dihydroxyindole-2-carboxylic acid (DHICA). The presence of Zn2+ and Cu2+ ions leads to relatively higher conversion into DHICA than into DHI [112]. The ratio between DHI and DHICA has a large impact on the properties of the pigments because DHICA absorbs less visible light than DHI and the antioxidant properties of DHICA are much higher [113]. In vivo, most dopachrome is converted to DHICA [111]. DHI and DHICA are converted to eumelanin. Eumelanin is a highly compact pigment, packed in eumelanosomes with a highly ordered glycoprotein matrix [114]. Eumelanin absorbs almost the full light spectrum and is perceived as a dark brown to black colour. People with brown eyes have relatively more eumelanin in comparison to pheomelanin.

When the *MC1R* gene is mutated, less cAMP is produced and less tyrosinase is activated. Without the presence of cAMP and tyrosinase, cysteine is incorporated, and the eumelanin pathway switches to the pheomelanin pathway [115]. The incorporation of cysteine results in the formation of 5-S-cysteinyldopa (5-SCD) and 2-S-cysteinyldopa (5-SCD) in a ratio of 5:1 [116]. 5-SCD and 2-SCD are rearranged in both 2H-1,4-benzothiazine (BTZ) and its 3-carbocyl acid (BTCZA). BTCZA is the carboxylated variant of BTZ and the presence of Zn2+ ions favours the conversion of BTCZA [117]. BTZCA absorbs visible light and UVA more efficiently, which results in the light-dependent synthesis of ROS [113]. BTZ is a stronger pro-oxidant and induces ROS production by reduction-oxidation (redox) cycling in the dark. There are multiple explanations for the feature of carboxylated molecules: they have a lower number of reactive sites, a lower oxidation potential and a negative charge, which affects the structure of the molecule and the susceptibility to post-synthetic modifications [113]. Both BTZ and BTZCA can be converted to pheomelanin. Pheomelanin is packed in smaller pheomelanosomes that consist of a loosely aggregated and disordered glycoprotein matrix [114]. This type of melanin reflects a much broader light spectrum than does eumelanin, which is perceived as a yellow to red colour. People with a lighter eye colour have relatively more pheomelanin than eumelanin [99,114].

Both types of melanin can absorb free radicals and inhibit UV-mediated damage [118,119]. However, pheomelanin is more phototoxic than eumelanin, because pheomelanin generates more ROS than eumelanin when exposed to light [120,121]. Pheomelanin has a lower ionization potential with a photoionization threshold of 326 nm, which falls in the UVA region. The photoionization threshold for eumelanin is 282 nm, which lies in the UVB region [122]. Exposure of pheomelanin to UVA not only leads to more production of ROS, but also to the oxidation of glutathione (GSH) and other oxidants [123,124]. The decrease in antioxidants can indirectly damage the DNA. As most UVB radiation is already absorbed and radiation in the UVA region is not, and the ionization potential of pheomelanin is lower; this may explain why a light iris colour is a risk factor for the development of a UM.

However, pheomelanin probably also has a UV-independent carcinogenic contribution. In a study on the development of cutaneous melanoma, a *BRAF* mutation was introduced in three different types of mice: red mice with an abundance of pheomelanin, albino mice with no melanin and black mice with an abundance of eumelanin [36]. When raised in the dark, around 50% of the red mice developed a cutaneous melanoma, while the development of cutaneous melanoma in the albino and black mice was sporadic. The damage to DNA and lipids was significantly higher in red mice in comparison to the albino mice.

Albinos lack pigmentation in their eyes, both in the iris as well as in the RPE, leading to foveal hypoplasia and decreased visual acuity, as well as in the iris, which leads to iris translucency. The iris contains some blood vessels, which are only visible when there is (almost) no melanin in the iris pigment epithelium [99].

UM itself can be non-pigmented, lightly or strongly pigmented, and it can be a combination of various degrees of pigmentation, e.g., a combination of both a non- and strongly-pigmented tumour segment. Whereas the melanosomes in melanotic melanoma produce both eumelanin and pheomelanin, amelanotic melanoma only produces pheomelanin [125,126]. The tyrosinase from melanosomes of amelanotic melanomas is less active [127].

## 10. Genes and Iris Colour

The two main genes that determine iris colour are ocular albinism II (*OCA2*) and HECT domain and RCC1-like domain 2 (*HERC2*), both of which are located on chromosome 15 [128,129]. People with blue eyes often have specific single nucleotide polymorphisms (SNPs) in *HERC2* and *OCA2*. *OCA2* encodes the P protein, which affects the amount and type of melanin in melanocytes. The specific SNP in the *OCA2* gene, which is associated with blue eyes, causes a reduction in *OCA2* transcription of that allele compared to the normal other allele. This causes an accumulation of tyrosinase, which leads to a defective eumelanin synthesis but no change in the synthesis of pheomelanin [126,130]. *HERC2* regulates OCA2 expression. Therefore, people with the blue-eye SNP in both genes have blue eyes. This proves that blue eyes must be mostly recessive. When someone has inherited the ”blue version” of *OCA2* from one parent, this gene makes less of the precursor of melanin. However, when someone gets the “‘brown version” of *OCA2* from the other parent, this *OCA2* polymorphism is turned on and will make melanin, so the child will have brown eyes [131]. *OCA2* and *HERC2* are probably the most important, but not the only genes that determine the amount of melanin and the eumelanin/pheomelanin ratio [132]. In 2009, scientists of the Erasmus University in Rotterdam, The Netherlands, developed an algorithm to predict iris colour, using DNA markers (The IrisPlex System). They found six SNPs that functioned as major genetic predictors of eye colour on six different genes: in addition to the already mentioned *HERC2* rs12913832 and *OCA2* rs1800407, these were *SLC24A4* rs12896399, *SLC45A2* rs16891982, *TYR* rs1393350 and *IRF4* rs12203592. Using these six genetic predictors of iris colour, the area under the curve (AUC) was 0.93 for brown, 0.91 for blue and 0.72 for intermediate coloured eyes, from which one may conclude that these six genes predominantly determine iris colour in humans [133]. Interestingly, MC1R is not one of the genes used in the IrisPlex algorithm, while it is used to predict hair colour.

In another study, several pigment genes that have previously been associated with cutaneous melanoma were identified as risk factors to develop UM [134]. Of the 28 SNPs that had already been identified as risk factors for cutaneous melanoma, three had already been linked to iris colour: SNPs rs12913832, rs1129038 and rs916977, located on the pigmentation genes *HERC2*, *OCA2* and *IRF4*, respectively. This study showed that specific alleles of these genes were associated with UM risk. No association between these pigmentation genes and UM tumour characteristics has as yet been described.

## 11. History of the Distribution of Eye Colour and Skin Colour in the World

A light eye colour is especially present in northern Europe and countries with immigrants from northern Europe. Initially, humans had brown eyes, and the genes for a certain type of eye colour could only increase when a group of people split off from its parent population. According to Frost, sexual selection is the reason for the wide-scale spreading of the blue eye colour [135]: men selected the women who stood out from the crowd and had the rarest eye colour. This phenomenon has also been described for animals [136]. A fair eye colour was the rarest colour at that moment. In this way, fair eye colours in Europe, western Asia and the Middle East could multiply.

Another theory for the spread of fair-skinned people with a light eye colour to northern Europe has to do with vitamin D. Primates have fair skin under their hairy fur. Similar to a dark skin, this kind of fur protects the skin from sunlight. About 65,000 years ago when a group of people moved from Africa to Asia and Europe, some developed a fair skin, like the primates used to have. Sun exposure is necessary for vitamin D, and, while pheomelanin allows more UV radiation to penetrate the skin, eumelanin absorbs UV radiation. People with more pheomelanin compared to eumelanin have more efficient vitamin D synthesis. A lack of vitamin D may lead to diseases such as rickets, osteomalacia and osteoporosis [137,138]. Furthermore, vitamin D is important for the immune system. Both the cells of the innate and the adaptive immune system have vitamin D receptors [139,140]. Therefore, in northern Europe, people with a fair skin would be in a better condition to live to an adult age and have children.

## 12. Conclusions

The determining factor of iris colour is the amount and type of melanin present in the iris melanocytes. The importance of vitamin D, perhaps in conjunction with the appeal of a distinctive eye colour, may have led to the selection of people with a light skin together with light eye colour, leading to a relatively high percentage of people with lighter eyes in northern Europe. Over time, this led to 78% of the Dutch population having light eyes.

All of the described data show that the risk of getting UM is higher when a person has lighter eyes (blue/grey or green/hazel): the frequency of UM is higher in light-eyed people than in dark-eyed people, and this is even noticeable among a group of people of mainly Caucasian ancestry. In particular, people with green/hazel iris colour seem to be at higher risk for developing UM. The cause of this greater risk is probably related to the type and ratio of melanin. Light-eyed people do not have a lot of eumelanin in their melanocytes but carry more pheomelanin. The harmful wavelength is probably not UV light, as these waves do not reach the part of the eye where most UM develops, i.e., the choroid and ciliary body. Most UV rays are absorbed by the cornea, the lens and the vitreous. However, other wavelengths such as visible light do penetrate the back of the eye and can be responsible for the production of toxic ROS, especially by pheomelanin. Pheomelanin by itself may also increase genotoxic damage, accumulated over time. We hypothesize that damage from environmental factors or age-related factors create more DNA damage in melanocytes that contain mainly pheomelanin than in melanocytes with mainly eumelanin. The degree of pigmentation of the iris reflects the type of melanocyte in the whole uvea. The present findings lead to a follow-up question: do UM in eyes with a light iris differ in behaviour from UM in eyes with a dark iris? The difference in the type of melanin may not only affect the chance of developing a UM, but also the tumour’s behaviour.

## Figures and Tables

**Table 1 ijms-21-07172-t001:** Baseline characteristics of the study population of the Rotterdam Study (RS123) population (n = 5951) and the Leiden Uveal Melanoma cohort (n = 412).

	Leiden Cohort (n = 412)		RS123 (n = 5951)		*p*-Value	Odds Ratio	95% CI
	*n*	% *	*n*	% *			
**Gender**					<0.001 ^†^		
Male	239	58%	2558	43%			
Female	173	42%	3393	57%			
**Age**							
Mean (SD)	63.0 (13.7)		66.4				
RS1			74.0 (8.2)				
RS2			67.7 (7.4)				
RS3			56.2 (5.8)				
**Iris Colour**					<0.001 ^‡^		
Brown	64	16%	1356	23%		1.0	
Green/hazel	79	19%	460	8%		3.64	2.57–5.14
Blue/grey	269	65%	4135	70%		1.38	1.04–1.82

* Percentages are rounded and may not total 100; ^†^ Pearson Chi Square Test; ^‡^ Fisher’s Exact Test.

**Table 2 ijms-21-07172-t002:** Iris colour distribution of Uveal Melanoma cases and controls from other studies.

Study	Iris Colour Cases *				Iris Colour Controls *			
	*Blue/Grey*	*Green/Hazel*	*Brown*	*n*	*Blue/Grey*	*Green/Hazel*	*Brown*	*n*
**Rootman et al., 1984 [15]**	72%	16%	12%	85	47%	28%	26%	687
**Gallagher et al., 1985 [16]**	66%	18%	15%	65	43%	26%	31%	65
**Pane et al., 2000 [17]**	42%	37%	21%	125	46%	29%	25%	373
**Guénel et al., 2001 [18]**	56%	22%	22%	50	34%	28%	39%	476
**Vajdic et al., 2001 [19]**	50%	37%	13%	246	45%	33%	22%	893
**Stang et al., 2003 [20]**	75%	14%	10%	118	64%	11%	24%	475
**Schmidt-Pokrzywniak et al., 2009 [21]**	67%	20%	14%	459	61%	17%	23%	827
***Total***	703 (61%)	283 (25%)	162 (14%)	1148	1889 (50%)	930 (25%)	973 (26%)	3796

* Percentages are rounded and may not total 100.

**Table 3 ijms-21-07172-t003:** Odds ratio to develop Uveal Melanoma in iris colour categories in other studies.

	Location	Type of Tumour	Odds Ratio Brown	Odds Ratio Green/Hazel	95% CI	Odds Ratio Blue/Grey	95% CI
**Rootman et al., 1984 [15]**	Canada	Uveal Melanoma	1	1.3	0.6–3.0	3.4	1.7–6.8
**Gallagher et al., 1985 [16]**	Canada	Ocular Melanoma	1	1.4	0.5–4.1	3.1	1.3–7.5
**Pane et al., 2000 [17]**	Australia	Ocular Melanoma	1	1.5	0.9–2.6	1.1	0.7–1.9
**Guénel et al., 2001 [18]**	France	Ocular Melanoma	1	1.4	0.6–3.4	2.9	1.4–6.1
**Vajdic et al., 2001 [19]**	Australia	Choroidal and Ciliary Melanoma	1	2.0	1.3–3.1	1.9	1.3–3.0
**Stang et al., 2003 [20]**	Germany	Uveal Melanoma	1	3.1	1.4–7.0	2.8	1.5–5.3
**Schmidt-Pokrzywniak et al., 2009 [21]**	Germany	Uveal Melanoma	1	2.0	1.4–3.0	1.8	1.3–2.5

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
