# Peer review of "Iris Colour and the Risk of Developing Uveal Melanoma"

_ijms, 2020, doi:10.3390/ijms21197172_

Round 1

Reviewer 1 Report

Houtzagers et al. studied the distribution of iris color between Dutch UM patients and healthy Dutch controls, and compared their results with the literature about iris color and risk of UM. They also discussed the pathological role of melanin in the development of UM. This review is well organized and easy to read.

  • The section “History of the Distribution of Eye Colour and Skin Colour in the World » could be remove from the review if no link is made with UM and iris color.
  • Lines 249-250: BRAF mutations are found in posterior UMs. See these publications: Janssen et al. 2008 (PMID: 18985043) and Goh et al. 2020 (PMID: 32305056).

Minor comments:

  • Please refer to Tables 2 and 3 within the text.
  • Missing references about iris color in UM: Regan et al 1999 (PMID: 10369595); Jensen 1963 (PMID: 14050563). Maybe some data could be included in Tables 2/3.
  • Missing references about UV light and UM development: Mallet et al. 2014 (PMID: 23981010); Goh et al. 2020 (PMID: 32305056).
  • Missing reference about non-UV wavelengths and UM development: Logan et al. 2015 (PMID: 26075084).
  • Please define abbreviations the first time they appear (ex. mutations or protein names at lines 164-170).
  • Line 216: replace “GNAQ A626C” by “GNAQ Q209P”.
  • Line 251: replace “BRAF1” by “BRAF”.
  • Line 255: replace “UVR radiation” by “UV radiation”.
  • Line 259: correct for “In order to reach the choroid, light has to penetrate multiple tissues [and humors]: the cornea, [the aqueous humor,] the lens, the vitreous [humor] and the retina. »
  • Lines 277-278: add references (for cell culture of pigmented ocular cells).
  • Lines 288-289: add references (for melanin types).
  • Lines 304-305: add references (for no expression of MC1R in uveal melanocytes).

Reviewer 2 Report

An interesting manuscript , re emphasising inferences from previous publications

However, the authors have not clearly demonstrated a direct observational cause-effect relationship and pathogenesis.

Will also be useful to comment on parallels in more pigmented races where light irides are occasionally found
